# Infectious Agents Identified by Real-Time PCR, Serology and Bacteriology in Blood and Peritoneal Exudate Samples of Cows Affected by Parietal Fibrinous Peritonitis after Caesarean Section

**DOI:** 10.3390/vetsci7030134

**Published:** 2020-09-13

**Authors:** Salem Djebala, Julien Evrard, Fabien Gregoire, Damien Thiry, Calixte Bayrou, Nassim Moula, Arnaud Sartelet, Philippe Bossaert

**Affiliations:** 1Clinical Department of Ruminants, University of Liège, Quartier Vallée 2, Avenue de Cureghem 7A-7D, 4000 Liège, Belgium; Calixte.Bayrou@uliege.be (C.B.); asartelet@uliege.be (A.S.); p.bossaert@uliege.be (P.B.); 2Gestion et Prévention de Santé, Regional Association of Health and Animal Identification, Allée des Artisans 2, 5590 Ciney, Belgium; julien.evrard@arsia.be (J.E.); fabien.gregoire@arsia.be (F.G.); 3Bacteriology, Department of Infectious and Parasitic Diseases, University of Liège, Quartier Vallée 2, Avenue Cureghem 6, B-4000 Liège, Belgium; damien.thiry@uliege.be; 4Department of Animal Production, University of Liege, Quartier Vallée 2, Avenue de Cureghem 6, 4000 Liège, Belgium; nassim.moula@uliege.be

**Keywords:** Parietal fibrinous peritonitis, caesarean section, peritoneal fluids, *Coxiella burnetii*, *Bovine Herpesvirus 4*, *Mycoplasma bovis*, *Trueperella pyogenes*, *Escherichia coli*

## Abstract

The aim of this study was to identify the pathogens potentially involved in parietal fibrinous peritonitis (PFP). PFP is a complication of laparotomy in cattle, characterized by an accumulation of exudate inside a fibrinous capsule. We have studied 72 cases of PFP in Belgian blue cows, confirmed by a standard diagnostic protocol. Blood was collected to evaluate the presence of antibodies for *Mycoplasma bovis*
*(M. bovis)*, *Coxiella burnetii*
*(C. burnetii)* and *Bovine Herpesvirus 4*
*(BoHV4*) by enzyme-linked immunosorbent assays. Peritoneal exudate was obtained from the PFP cavity to perform bacteriological culture, and to identify the DNA of *M*. *bovis*, *C*. *burnetii* and *BoHV4* using real time polymerase chain reaction (qPCR). Bacteriological culture was positive in most peritoneal samples (59/72); *Trueperella pyogenes* (*T. pyogenes*) (51/72) and *Escherichia coli* (*E. coli*) (20/72) were the most frequently identified. For *BoHV4*, the majority of cows showed positive serology and qPCR (56/72 and 49/72, respectively). Contrariwise, *M. bovis* (17/72 and 6/72, respectively) and *C. burnetii* (15/72 and 6/72, respectively) were less frequently detected (*p* < 0.0001). Our study proves that PFP can no longer be qualified as a sterile inflammation. Moreover, we herein describe the first identification of *BoHV4* and *C. burnetii* in cows affected by PFP.

## 1. Introduction 

Parietal fibrinous peritonitis (PFP) in cattle is a postoperative complication of laparotomy [1,2,3], characterized by the accumulation of fibrin and peritoneal exudate inside a thick fibrous capsule between the outer sheath of the parietal peritoneum and the abdominal muscular layers [1,3]. Symptoms of PFP occur several weeks after surgery, and may include hyperthermia, anorexia, weight loss, visual abdominal distention, and colic [3,4,5]. 

In Belgium, PFP is frequently encountered in rural veterinary practice due to the large number of elective caesarean sections (CS) performed in the Belgian Blue breed [6]. Its incidence after CS has been estimated to be 1%, and its mortality has been estimated at 13% [7,8]. Unfortunately, PFP is very scarcely documented and rural practitioners have little information considering its treatment, prevention and prognosis. In particular, the aetiology of PFP is the subject of speculation. For a long time, PFP has been considered as an aseptic inflammation [5,8,9]. This assumption has recently been challenged, after the isolation of several aerobic and anaerobic bacteria such as *Trueperella pyogenes* (*T. pyogenes*), *Escherichia coli* (*E. coli*), Staphylococcus aureus, Pseudomonas aeruginosa, Proteus mirabilis, Fusobacter necrofurum, Comamonas kerstersii, Bacillus licheniformis, and Bacteroides species in in the peritoneal fluid of PFP cows [2,3,4]. 

The last decade, three infectious agents have received ample attention in Belgian rural practice, i.e., *Mycoplasma bovis* (*M. bovis*), *Coxiella burnetii* (*C. burnetii*) and *Bovine Herpesvirus 4* (*BoHV4*). All three germs have been identified in various bovine infectious disorders including reproductive tract disease, abortion, mastitis, respiratory diseases, and arthritis [2,10,11,12], and the number of positive laboratory diagnoses shows an increasing trend [13,14]. Their implication in peritonitis and PFP is unclear.

The aim of this study was to perform bacteriological culture on peritoneal fluid samples of a large cohort of cows presenting PFP. Furthermore, to gain a broader insight into the aetiology of PFP, we aimed to evaluate the implication of *M. bovis*, *C. burnetii* and *BoHV4* in PFP, by determining the presence of antibodies in the blood and genetic material in peritoneal fluid samples from cows presenting PFP. 

## 2. Material and Methods

Between March 2017 and March 2018, the Clinical Department of Production Animals (University of Liege in Belgium) and the Regional Association of Animal Health and Identification (ARSIA) collaborated with Belgian rural veterinary practitioners in a project to obtain diagnostic elements on PFP in Belgian blue cattle breed. All rural veterinarians from the ARSIA database were contacted by e-mail, and instructions for the diagnosis and treatment of PFP were published online. In each case suspected of PFP based on clinical signs and ultrasound findings, an aseptical paracentesis of the cavity was carried out to confirm the diagnosis, as described in previous studies [3,4]. Finally, PFP was confirmed in 72 cases, and 10 mL of peritoneal fluid was collected in each case for further examination. Furthermore, blood samples were obtained from the coccygeal vein using non-coagulant Vacutainer^®^ tubes (BD, Plymouth, UK) for biochemical analysis, and an additional blood tube was collected for further examination of the serological status of the animals. Finally, the treatment (and definitive diagnosis) consisted of surgical draining of the PFP cavity. All blood and peritoneal fluid samples were kept at 4 °C and dispatched to the ARSIA laboratory. The national identification database (SANITEL) was consulted afterwards for the cattle age. Consent was obtained from all veterinarians and owners to use the samples to perform the current study and publish the results. 

All invasive procedures (paracentesis, blood sampling and surgical drainage) were done in cases encountered in the field, primarily for diagnostic and therapeutic purposes. At no point did the research protocol interfere with treatment decisions and housing or management of the cows. Therefore, the animals in our study did not fall into the definition of an experimental animal, and no ethical approval was required.

Peritoneal exudate samples were used for aerobic and anaerobic bacteriological culture and for the detection of *C. burnetii*, *M. bovis* and *BoHV4* genetic material. Blood samples were used for the detection of *C. burnetii*, *M. bovis* and *BoHV4* antibodies. 

The samples for aerobic culture were grown on Columbia agar, Gassner and Columbia/Nalidixic acid agar media (Thermo Fisher Scientific, Brussels, Belgium) at 37 ± 2 °C. Samples for anaerobic culture were grown under anaerobic conditions on Schaedler medium (Thermo Fisher Scientific, Brussels, Belgium) at 37 ± 2 °C. Two readings of each medium were performed at 18 to 24 h and 36 to 48 h of incubation. Bacterial identification was performed by the Maldi Biotyper^®^ (Bruker Daltonics, Bremen, Germany). The culture was considered “negative” if no bacterial growth was observed, and “positive” when one or several bacteria were found.

The detection of *M. bovis*, *C. burnetii* and *BoHV4* antibodies was performed using commercially available enzyme-linked immunosorbent assay (ELISA) kits: Monoscreen AbELISA *BoHV-4* indirect bicupule (Bio K263)^®^ (BioX, Rochefort, Belgium), Monoscreen AbELISA *Mycoplasma bovis* indirect monocupule (Bio K302)^®^, (BioX, Rochefort, Belgium) and PrioCHECK™ Ruminant Q Fever Ab Plate Kit (ELISACOXLS)^®^ (Thermo Fisher Scientific, Rochefort, Belgium).

The ELISA test results of *BoHV4* and *C. burnetii* are semi-quantitative. The antibody concentration (%) is calculated as the ratio between the optic density of the tested sample and a control sample, multiplied by 100. Results for *BoHV4* and *C. burnetii* were classified as “negative” (relative density below 30% and 40%, respectively), as “positive” (relative density between 30% and 120% and between 40% and 300%, respectively) or as “highly positive” (relative density above 120% and above 300%*,* respectively). The results for the *M. bovis* ELISA kit are only qualitative (“positive” or “negative”).

The presence of genetic material of *M. bovis*, *C. burnetii* and *BoHV4* was analyzed in peritoneal fluids samples using real time polymerase chain reaction (qPCR). The DNA extraction was achieved using MagAttract 96 cador Pathogen Kit^®^ (QIAGEN, Antwerp, Belgium) and an extraction robot KingFisher™ Flex 96^®^ (Thermo Fisher Scientific, Brussels, Belgium), according to the manufacturer’s instructions. Three commercially available kits were used: LSI VetMAX Bovine Herpes Virus Type 4^®^ (Thermo Fisher Scientific, Brussels, Belgium), LSI VetMAX Mycoplasma bovis^®^ (Thermo Fisher Scientific, Brussels, Belgium) and LSI VetMAX Coxiella burneti-Absolute Quantification^®^ (Thermo Fisher Scientific, Brussels, Belgium). Thermal cycling conditions were controlled using Thermocycleur ABI7500^®^ (Thermo Fisher Scientific, Brussels, Belgium). A “negative”, “positive” or “highly positive” result was obtained, corresponding to replication cycles (Ct) below 45, between 45 and 30, or below 30, respectively.

Statistical analyses were performed using SAS (2001). Descriptive analysis was carried out for the age of cows and the number of bacteria cultured in the peritoneal exudate. Continuous data (age, number of bacteria found by bacteriology) were checked for normal distribution with a Shapiro–Wilk test, and displayed as the median and range in case of non-normal distribution. Chi-square and Fisher tests were used to compare the number of positive and negative results of bacteriological culture, ELISA and qPCR, and to compare the germ-specific proportions within positive samples of bacteriology, ELISA and qPCR. Moreover, a Chi-square test was used to compare the antibody and DNA concentration in the positive, semiquantitative samples. A test of independence was performed to evaluate the relation between ELISA and qPCR outcomes and the results of bacterial culture. The procedure “Proc Freq” in SAS was used for all statistical analyses; the cut-off of significance was fixed at *p* < 0.05.

## 3. Results 

In total, 37 rural veterinarians collected blood and peritoneal exudate samples from 72 cows affected by PFP after CS in 61 Walloon farms. The age of cows affected by PFP varied from 26 to 120 months with a median of 45 months. 

Bacteriology was positive in the majority (59/72) of cows and negative in only 13/72 samples (*p* < 0.0001). The number of bacteria identified in the positives samples varied between 1 to 3 with a median of 1. 

In total, 82 different strains from nine bacteria species were identified in the positive samples, among which *T. pyogenes* (51/59) and *E. coli* (20/59) were predominant compared to the other sporadically identified bacterial species (*p* < in the remaining samples). *E. coli* was identified alone in 4/20 samples and in association with other bacteria in the other 16/20 peritoneal samples, especially *T. pyogenes*. The other isolated bacteria species were always associated with *T. pyogenes* or *E. coli*, except for *Helicoccus ovis* and *Streptococcus mitis*, which were identified alone. Aerobic bacteria were more frequently identified than anaerobic (*p* < 0.0001); at least one aerobic bacteria (mainly *T. pyogenes* and *E. coli*) was cultured in all positive samples (59/59), while anaerobic bacteria (exclusively *Clostridium perfringens*) were observed in only 4/59 samples and were always associated with aerobic bacteria. All the results of bacteriological culture are summarized in the Scheme 1.

Antibodies against at least one of three germs were detected in the majority of blood samples (61/72); only 11/72 were fully negative (*p* < 0.0001). Antibodies of *BoHV4* were the most frequently detected (56/72), compared to those of *C. burnetii* (15/72) and *M. bovis* (17/72) (*p* < 0.0001). The majority of positive *C. burnetii* (14/15) and *M. bovis* (13/17) serology results were associated with a positive *BoHV4* ELISA. Within the positive samples, the concentration of *BoHV4* antibodies was “highly positive” in most cases (53/56), and “positive” for the other samples (3/56) (*p* = 0.002). Within positive ELISA results for *C. burnetii*, concentrations were “positive” in all cases, and never “highly positive”. Details of the ELISA results are displayed in Scheme 2.

The details of qPCR results for the different germs are displayed in Scheme 2. The qPCR analysis for *BoHV4*, *C. burnetii* and *M. bovis* confirmed the genetic material of at least one germ in the majority of samples (52/72), while 20/72 were completely negative (*p* = 0.0001). Genetic material of *BoHV4* was identified most frequently in peritoneal exudate (49/72) compared to *C. burnetii* (6/72) and *M. bovis* (6/72) (*p* < 0.0001). Furthermore, *C. burnetii* (3/6) and *M. bovis* DNA (6/6) were in most cases found in combination with *BoHV4.* The quantity of *BoHV4* genetic material was “highly positive” in most peritoneal samples (30/49), and “positive” in the other positive samples (19/49) (*p* = 0.116). The level of “highly positive” was reached in none of the positive *C. burnetii* samples, and in only 1/6 *M. bovis* samples. 

For *BoHV4,* the results of ELISA corresponded to those of qPCR in the majority of cases. In other words, in most cows having a negative serology for *BoHV4*, a negative qPCR was found, and the positive qPCR results corresponded with a positive serology. In rare cases, a positive ELISA result was found in combination with a negative qPCR, or vice versa. For *C. burnetii* and *M. Bovis*, a negative result for qPCR and ELISA was observed in the majority of cases. In contrast to *BoHV4*, several discrepancies between ELISA and qPCR were found; the majority of ELISA positive samples were negative to qPCR, while around half of qPCR positive samples were ELISA negative. The combination of qPCR and ELISA results for the three targeted germs is displayed in Table 1. 

A positive statistical association was found between the qPCR and ELISA results for *BoHV4* and bacteriological culture results. This relation between qPCR and ELISA results and bacteriological culture results could not be confirmed in the case of *C. burnetii* or *M. bovis*. The combinations of qPCR and ELISA results and bacteriology outcomes are displayed in Scheme 3.

## 4. Discussion 

This study presents a unique dataset containing a large number of PFP cases observed in the field. Only very few studies have reported aerobic or anaerobic bacteria, *BoHV4* and *M. bovis* in peritoneal liquids and, particularly, in cases of PFP [2,3,4,15]. To our knowledge, the presence of *C. burnetii* in peritoneal fluids has never been demonstrated.

At least one bacterial species was cultured in more than 80% of the tested samples in this study. This number may even be an underestimation of the true presence, due to the limited sensitivity of bacteriological culture. Also, it is very likely that several cows had been treated with antimicrobials before sampling, modifying the culture results [2,4,16,17].

In a previous publication, anaerobic bacteria have been mainly isolated from the peritoneum during CS [18]. This led to the assumption that bacteria originating from the endogenous vaginal flora and the incised uterus were the main contaminants leading to infectious complications after CS. In the current study, in contrast, aerobic bacteria were isolated far more frequently from PFP than anaerobic germs. *T. pyogenes* and *E. coli* were predominant, confirming earlier reports on PFP and generalized peritonitis [3]. 

*T. pyogenes* and *E. coli* are ubiquitous in the environment [19,20] and colonize a wide range of tissues and organs [20,21,22]. Hence, it seems logical that exogenous contamination by *T. pyogenes* and *E. coli* during CS is a primary cause of infectious complications, including PFP and peritonitis [23,24,25]. Evidently, the risk of complications increases in the case of a massive contamination or immunosuppression [22,26,27]. On the other hand, since healthy cows can have a physiological bacteraemia [21], it is also plausible that PFP is the result of secondary haematogenous infection of a sterile fluid-filled cavity. In conclusion, PFP is in the majority of cases bacterially contaminated, but it remains to be elucidated whether bacteria are primary agents of PFP originating from the environment, the surgeon’s hands, the surgical material or the cow’s skin or organs [19,20,22], or secondary contaminants of an initially sterile process. 

A large number of cows suffering from PFP displayed a positive serology and/or qPCR for one or more of three emerging pathogens, i.e., *BoHV4, C. burnetti* and *M. bovis*. These three germs can invade multiple tissues [13] and share the potential to invade white blood cells, allowing them to escape the host’s immune response and to pass into a dormant phase [28,29,30]. Stress, parturition and inflammatory processes can reactivate a dormant infection [2,29,31]. Hence, their presence in the PFP fluids may be the result of haematogenous spread via immune cells toward an inflammatory site, and their true implication in the pathogenesis of PFP remains to be elucidated.

The combination of serology and qPCR results allows some interpretation. Animals with an active infection will typically have a positive serology and high amounts of DNA in the infection site [14,32,33]. A positive qPCR in the absence of blood antibodies is indicative of a recent infection; the time between primary infection and detectable antibody levels ranges from 10 days to 4 weeks for the three germs [29,32,34]. A negative qPCR in the presence of antibodies indicates an inactive infection, inhibition of reactivation by a serologic response [29,33,34], or intermittent bacterial replication [35]. There is a long persistence of antibodies in blood circulation after primary infection [2,36].

Of the three studied germs, *BoHV4* yielded the most positive results: over 75% of blood samples were seropositive. This is in line with the endemic situation of this virus in Belgian herds, particularly in beef cattle and older cows [14,37]. For comparison, the seroprevalence has been reported to be 67.5% in aborting cows in Wallonia [14]. Nearly 70% of peritoneal fluids were highly positive for qPCR, mostly in association with a strongly positive serology, indicating an active or reactivated infection. A negative qPCR in combination with a (highly) positive ELISA result was observed in a small number of cases, indicating latency of the virus and a serological response. Since a direct relation between *BoHV4* detection and specific lesions has never been established [38], the relevance of *BoHV4* in the pathogenesis of PFP remains unclear. 

Antibody or DNA detection for *C. burnetii* and *M. bovis* yielded far fewer positive results than *BoHV4*. *C. burnetii* and *M. bovis* are common in Belgium: 57.8% of tested herds in Wallonia have been reported to have seropositive animals for *C. burnetti*, and 30% of herds contain animals actively excreting the germ [33]. Between 2012 and 2016, the apparent herd seroprevalence for *M. bovis* has been estimated to be 23.6% [13]. The combination of a negative serology and a negative qPCR in peritoneal fluid was most frequently found. A positive ELISA in combination with a negative qPCR was detected in a number of cases, indicating an inactive infection or a serological inhibition of reactivation [29,33]. As for *M. bovis,* this may also be due to the intermittent bacterial replication [35].

A positive association was observed between the presence of *BoHV4* (DNA and/or antibodies) and the bacteriological results. Moreover, *C. burnetii* and *M. bovis* antibodies and DNA were rarely found alone, as reported elsewhere [36,39], and were almost always associated with *BoHV4* antibodies and DNA. It can be postulated that a decrease of immunity induced by a *BoHV4* infection may increase the risk of bacterial co-infection [28,38,40,41]. Inversely, the inflammatory condition caused by bacterial infection might also induce *BoHV4* reactivation [40,41].

It should be stressed that the presence of germs, their genetic material or their antibodies in PFP cows does not prove a causal mechanism. Their exact role in the pathogenesis of PFP requires further studies. The presence of peritoneal fluids in matched negative control cows could have shed more light on the importance of a positive bacteriology or qPCR, but this was not feasible in the current study setup. 

## 5. Conclusions 

PFP is a frequent pathology in Belgium. Our study clearly demonstrates that PFP can no longer be considered as a sterile process. Our study confirms previous reports of *M. bovis* in the peritoneal fluid of cows presenting PFP and adds the PFP as new target sites for *BoHV4*, *C. burnetii* and other bacterial species. These germs can colonize the PFP through endogenous and exogenous contaminations of CS or via haematogenous spread. Their exact role in the pathogenesis of PFP cannot be concluded from this dataset and requires further studies.

## Data Availability

The main data are presented in the paper. However, raw data files can be provided upon request.

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
