# Peer review of "Infectious Agents Identified by Real-Time PCR, Serology and Bacteriology in Blood and Peritoneal Exudate Samples of Cows Affected by Parietal Fibrinous Peritonitis after Caesarean Section"

_vetsci, 2020, doi:10.3390/vetsci7030134_

Round 1
Reviewer 1 Report
Interesting work about a disease which seems to gain importance on the Belgian field. I wondered about the relevance of certain conclusions concerning the results obtained, but the authors take sufficient precautions in the conclusion part about the real implication of their results that this suits me in the end: the implication of these results is still uncertain as long as the physiopathology is not elucidated.
Some questions and remarks :
- what is the belgian seroprevalence of BHV-4, Mycoplasma bovis and Coxiella burnetii in "non ill cattle" ? It's something that could be interesting to know in the discussion if ever it is known.
- line 129 : reference to the figure is missing
Author Response
Please see the attachement

Reviewer 2 Report
The manuscript reports performing different approaches to identify infectious bacteria in blood and peritoneal samples in cows had parietal fibrinous peritonitis (PFP) After caesarean section. This study is crucial since authors used a huge cohort of cows to challenge the paradigm about PFP as a sterile process. The manuscript is well-written and the methods section is clear. However, the following comments to the authors need to address before the manuscript can be considered for publication.
L19: Add the cow breed
L46: Delete “isolated”
L46: Provide examples for aerobic and anaerobic bacteria
Methods sections: Add the number of cows
L61: add “… in Belgium.”
L62: Add the cow breed
L102: Which SAS procedure(s) used in your analyses?
Scheme III: Add info for y-axis
Author Response
Please see the attachement

Reviewer 3 Report
It is a well designed study, with clear aims. There are adequate sampling procedures and laboratory examinations to support the aims of the study. Results are clearly presented and, in general, the text is well written, easily comprehensive without mistakes.
It seems that this is a part of a bigger work and here a portion is presented.
My main comments concern the material and method section, and especially the sample collection (L64-65)
It would be useful if the selection criteria for sampling were provided here. The website of the info could be provided, or the questionnaire about the clinical criteria of animal selection could be here as attachment.
Moreover it is not clear how the whole sampling procedure was performed. It is mentioned that (L65-66) “the 10 ml of peritoneal fluid were aseptically collected via paracentesis before surgical drainage”. So surgical drainage was performed in all animals? It was performed during the aforementioned (L64) exploratory laparotomy?
I would suggest that it would be useful to provide a clear step by step description of the animal selection, clinical examinations and sampling procedure including all manipulations in animals (e.g. ultrasound examination, paracentesis, laparotomy)
A grammar mistake; L46, one of isolation or isolated is enough
Author Response
Please see the attachement
